# Remimazolam’s Effects on Postoperative Nausea and Vomiting Are Similar to Those of Propofol after Laparoscopic Gynecological Surgery: A Randomized Controlled Trial

**DOI:** 10.3390/jcm12165402

**Published:** 2023-08-20

**Authors:** Ayumu Matsumoto, Shiho Satomi, Nami Kakuta, Soshi Narasaki, Yukari Toyota, Hirotsugu Miyoshi, Yousuke T. Horikawa, Noboru Saeki, Katsuya Tanaka, Yasuo M. Tsutsumi

**Affiliations:** 1Department of Anesthesiology and Critical Care, Hiroshima University, Hiroshima 734-8553, Japan; sennsinn720@gmail.com (A.M.); ssatomi@hiroshima-u.ac.jp (S.S.); ijaran7@gmail.com (S.N.); ykrtyt@gmail.com (Y.T.); h-miyoshi@hiroshima-u.ac.jp (H.M.); yousuke.horikawa@sharp.com (Y.T.H.); nsaeki@hiroshima-u.ac.jp (N.S.); 2Department of Anesthesiology, Tokushima University, Tokushima 770-8503, Japan; kakuta.nami@tokushima-u.ac.jp (N.K.); katsuya.tanaka@tokushima-u.ac.jp (K.T.); 3Department of Pediatrics, Sharp Rees-Stealy Medical Group, San Diego, CA 92123, USA

**Keywords:** remimazolam, PONV, general anesthesia

## Abstract

(1) Background: Remimazolam is a novel benzodiazepine that prevents postoperative nausea and vomiting (PONV), is more effective than volatile anesthetics, and was recently approved for use in Japan. (2) Methods: This prospective, double-blind, randomized controlled trial study aimed to compare the efficacy of remimazolam and propofol as general anesthetics in terms of the incidence of PONV after laparoscopic gynecological surgery (UMIN000046237). High-risk female patients who underwent general anesthesia with either remimazolam or propofol for the maintenance of anesthesia were enrolled. The primary outcome was the incidence of PONV in the two groups (i.e., REM versus PROP) 2 h and 24 h after surgery. The incidence of vomiting without nausea, rescue antiemetic use, and the severity of nausea were also evaluated. (3) Results: No significant differences in PONV were identified between the REM and PROP groups at 2 h or 24 h. Furthermore, no differences were observed in any of the measured parameters, and no adverse events were reported. (4) Conclusions: The results of the present study suggest that remimazolam may be as effective as propofol in preventing PONV; however, further investigation is necessary to identify possible differences between these two agents.

## 1. Introduction

Postoperative nausea and vomiting (PONV) constitute a complication of general anesthesia and occur in approximately 30% of patients who undergo general anesthesia for procedures [1,2], being undesirable for both patients and hospitals [3]. For patients, this is a significant factor that reduces satisfaction, similar to postoperative pain and intraoperative awakening [4,5]. In a previous survey study, patients ranked emesis as the most undesirable and nausea as the fourth most undesirable of 10 negative postoperative outcomes, while postoperative pain ranked third [6]. Although PONV is self-limiting and nonfatal in most cases, it can lead to bleeding, esophageal rupture, and life-threatening airway compromise [7,8]. PONV is also associated with a prolonged post-anesthesia care unit stay and unanticipated hospital admission, resulting in a significant increase in overall healthcare costs [3,9,10]. In the United States, the annual cost of PONV is estimated to be as high as USD 1 billion [11].

The risk factors for PONV include the following: female sex; age < 50 years; non-smoking status; surgical technique (laparoscopic, bariatric, gynecological, or cholecystectomy); a history of PONV/motion sickness; and opioid analgesia (including postoperative opioids). The protocols for prophylactic antiemetic use depend on the number of patient risk factors [12].

Remimazolam besylate (Mundipharma K.K., Tokyo, Japan) is a hypnotic sedative that was approved for general anesthesia in 2020 [13]. It has a chemical structure similar to that of midazolam and increases aminobutyric acid A receptor activity to induce cell membrane hyperpolarization, thereby inhibiting neural activity. Remimazolam is an ester-based benzodiazepine that is rapidly hydrolyzed, mainly by tissue carboxylesterases in the liver, to an inactive metabolite, with an approximately 300 times lower affinity than that of its parent compound [14,15]. The half-life of arterial remimazolam concentration for a 3 h constant-rate infusion is approximately 7.5 min [16]. Complications such as delayed emergence or re-sedation after flumazenil reversal have been reported in cases where it has been used for general anesthesia [17,18]. Previously, remimazolam was shown to have improved effects compared with desflurane, a volatile anesthetic, in reducing PONV [19]. Similarly, propofol demonstrated superior results compared to desflurane [20]. However, no studies have investigated the efficacy of remimazolam compared to propofol in preventing PONV. We hypothesized that propofol would be more effective than remimazolam in preventing PONV during laparoscopic gynecological surgery.

## 2. Materials and Methods

### 2.1. Patients and Study Protocol

This two-center, prospective, randomized controlled trial was approved by the Human Research Ethics Committee of Tokushima University Hospital (4119; Tokushima, Japan) and Hiroshima University Hospital (C-343; Hiroshima, Japan) and registered in a clinical trial database (UMIN000046237). Written informed consent was obtained from all the patients, and the study was conducted in accordance with the principles outlined in the Declaration of Helsinki.

Female patients > 20 and <80 years of age with an American Society of Anesthesiologists (ASA) physical status I–III who were scheduled to undergo laparoscopic gynecological surgery (hysterectomy, cystectomy, myomectomy, or sacrocolpopexy) under general anesthesia were enrolled in this prospective study between February and September 2022. Patients with an ASA physical status of IV or V, individuals who were pregnant or unable to provide consent, and those who were administered fentanyl or flumazenil during surgery were excluded from the study. Smoking status, history of PONV, and motion sickness were recorded for all patients before surgery.

Before surgery, the patients were randomly assigned in a double-blind manner to one of two groups using computer software (QuickCalcs, GraphPad Inc., La Jolla, CA, USA): the remimazolam (REM) group or propofol (PROP) group.

No premedications were administered. The standard monitoring included electrocardiography, noninvasive arterial blood pressure, pulse oximetry, capnography, train-of-four (ToF) electromyography, and an electroencephalogram (EEG) monitor (BIS^TM^, Medtronic Inc., Dublin, Ireland; Entropy, GE Healthcare, Chicago, IL, USA). In the PROP group, anesthesia was induced using remifentanil 0.3 μg/kg/min, propofol was induced using target-controlled infusion with an effect site concentration of 3 μg/mL, and rocuronium was induced with 0.6 mg/kg to facilitate endotracheal intubation and maintained with remifentanil and propofol. In the REM group, anesthesia was induced using remifentanil 0.3 μg/kg/min, remimazolam 12 mg/kg/h, and rocuronium 0.6 mg/kg to facilitate endotracheal intubation, and anesthesia was managed using remimazolam 0.4–1 mg/kg/h and remifentanil. In both groups, propofol, remifentanil, and remimazolam were administered at regulated doses to maintain an EEG value of 40–60. Similarly, rocuronium was administered when necessary for neuromuscular blockade.

For postoperative analgesia, peripheral nerve blocks (rectus sheath and transversus abdominis plane blocks) were performed using 60 mL of 0.25% ropivacaine after anesthesia induction or at the end of surgery. Acetaminophen 15 mg/kg and flurbiprofen 50 mg were administered at the time of wound closure. Intravenous ondansetron (4 mg) was administered at the end of surgery, muscle relaxation was reversed using sugammadex 2–4 mg/kg at the end of the surgery to confirm a ToF ratio > 90%, and the patient was extubated. In the REM group, flumazenil was not administered for reverse sedation. Hypotension was defined as a decrease in systolic blood pressure < 80% of the baseline obtained in the ward before surgery and was treated with 4–8 mg ephedrine. The postoperative rescue analgesics and antiemetics included pentazocine (15 mg) and/or buprenorphine (0.2 mg) and metoclopramide (10 mg) if the patient complained of pain or nausea. Fentanyl was not used as an intraoperative or postoperative analgesic. The patients were permitted to consume liquids 2 h after surgery and to consume solid food the following day.

### 2.2. Measurements

Anesthesiologists who were blinded to the patients’ information performed all the assessments and were different from those who administered the anesthesia. The primary outcome was the incidence of PONV in both groups. The secondary outcomes included vomiting without nausea, rescue antiemetic use, and nausea severity (nausea score: 0, no nausea; 1, mild nausea; 2, moderate nausea; 3, severe nausea).

All parameters were evaluated and recorded postoperatively at 2 h and 24 h, and any adverse events that occurred within 24 h postoperatively were recorded.

### 2.3. Sample Size and Statistical Analysis

The incidence of PONV after general anesthesia with sevoflurane and propofol has been reported to be 38.5% and 4.2%, respectively [21]. Therefore, we considered the 34% reduction in PONV with volatile anesthetics versus propofol to mark a significant difference. To identify the 34% reduction with α = 0.05 and a power of 80%, the required sample size was calculated to be 21 patients per group. Initially, 65 patients were included in this study, 60 of whom completed it. All statistical analyses were performed using Prism version 9 (GraphPad Inc., La Jolla, CA, USA). The two groups were compared using Fisher’s exact test, the unpaired Student’s *t*-test, chi-squared test, or Mann–Whitney U-test. Data are expressed as the mean ± standard deviation, and differences with *p* < 0.05 were considered to be statistically significant.

## 3. Results

Sixty-five women were enrolled in this study between February and September 2022, none of whom were initially excluded. In total, 33 and 32 patients were randomly assigned to the REM and PROP groups, respectively (Figure 1). In total, 5 patients were excluded from the study due to surgical complications, resulting in 30 patients in each group. There were no significant differences in patient background between the two groups (Table 1). No significant differences were found for all the examined endpoints, including the incidence of PONV, vomiting without nausea, rescue antiemetic or analgesic use, and nausea severity in the two groups.

There were no significant differences between the REM and PROP groups in terms of the incidence of PONV (30% versus (vs.) 30%, respectively; *p* > 0.05), rescue antiemetic use (1 vs. 5, respectively; *p* > 0.05), analgesic use, or the nausea score (*p* > 0.05). No parameters were significantly different at 24 h after surgery (Table 2). No adverse events were recorded during the study period and no patients were administered transdermal narcotics, steroids, or antiemetics.

## 4. Discussion

The incidence of PONV in the REM and PROP groups was similar at 2 h and 24 h after laparoscopic gynecological surgery. In addition, the antiemetic use and nausea scores of the two groups were similar. There were no significant differences in any of the measured parameters between the two groups. These results suggest that remimazolam and propofol were similarly effective in reducing PONV within the first 24 h after general anesthesia.

In 1999, Apfel identified four risk factors: female sex, a history of PONV and/or motion sickness, nonsmoking status, and the use of postoperative opioids. Collectively, these factors can predict the risk of developing PONV [1]. Other factors, such as surgical technique and anesthesia, also increase the risk of PONV [22]. Previous studies have attempted to reduce the incidence of PONV by optimizing medication use, the anesthesia methodology, and postoperative analgesia [23]. Previous opioid use, both intraoperatively and postoperatively, may increase the risk of PONV. The use of non-narcotic analgesics and local infiltration anesthesia without fentanyl in the perioperative period reduced the incidence of PONV [24]. Volatile anesthetics are known to increase the incidence of PONV. A comparison of general anesthesia with volatile anesthetics and propofol showed a decrease in PONV in the PROP group [25].

Interestingly, previous studies have reported that the use of midazolam reduces PONV [26]. REM is similar in chemical structure to midazolam and is believed to be as effective as midazolam in preventing PONV [27]. In a previous study, the incidence of PONV in surgery using propofol was 4.2–27.8% [21,28], and the probability of PONV using remimazolam was reported to be 3.7–27% [19,29], depending on the type of surgery and patient demographics. Several studies have compared the sedative effects and the postoperative quality of recovery using remimazolam and propofol [30,31]. In these studies, no significant differences were observed in the incidence of PONV. In our study, the incidence of PONV at 2 h and 24 h postoperatively was not significantly different between the REM and PROP groups. In both groups, the postoperative incidences of PONV at 2 h and 24 h were 30% and 17%, respectively. These results are similar to those reported in other studies, and the difference in the frequency of PONV may be due to the type of surgery and patient characteristics.

Our study had several limitations. First, the effect of remimazolam on PONV was only evaluated in high-risk female patients. Second, all patients were treated with ondansetron according to the standard care protocols. The strong antiemetic effect of ondansetron may have masked the differences between remimazolam and propofol. Finally, this was an investigational study with a small sample size. Additional studies with larger sample sizes are needed to investigate the effects of remimazolam on PONV between different sexes and types of surgery.

## 5. Conclusions

General anesthesia with remimazolam exhibited no significant difference from that with propofol, which suggests that there are similar antiemetic effects between these agents. Nevertheless, further investigation is necessary to identify possible differences between remimazolam and propofol in the incidence of PONV.

## Figures and Tables

**Figure 1 jcm-12-05402-f001:**
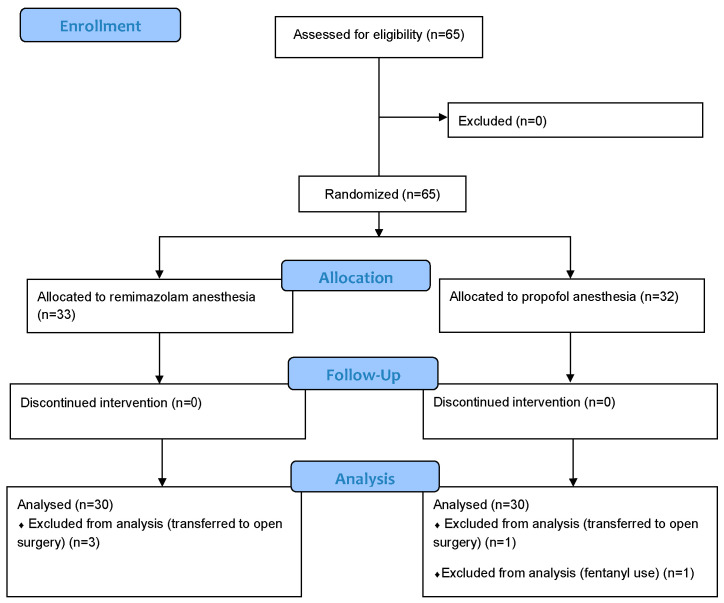
CONSORT 2010 flow diagram.

**Table 1 jcm-12-05402-t001:** Patient demographics.

	REM Group	PROP Group	*p* Value
	(n = 30)	(n = 30)	
Age, yr	50.0 ± 15.7	46.5 ± 16.9	0.42
Height, cm	157.0 ± 6.1	157.3 ± 5.3	0.81
Weight, kg	58.5 ± 9.5	60.9 ± 13.9	0.57
ASA PS I/II/III	9/21/0	11/19/0	0.3
PONV risk factor			
Tobacco use (n)	0	1	>0.99
History of motion sickness/PONV (n)	20	19	>0.99
Woman (n)	30	30	>0.99
Duration of anesthesia, min	211 ± 59	216 ± 46	0.77
Duration of surgery, mim	158 ± 55	151 ± 46	0.43
Fluid volume, mL	1199 ± 406	1201 ± 336	0.91
Intraoperative remifentanil (mg)	3.1 ± 1.3	2.6 ± 0.8	0.15

Data are presented as mean ± SD (range) or number of patients. PONV = postoperative nausea and vomiting.

**Table 2 jcm-12-05402-t002:** Postoperative complications.

	REM Group	PROP Group	*p* Value
	(n = 30)	(n = 30)	
2 h			
PONV	9 (30%)	9 (30%)	>0.99
Vomiting	3 (10%)	2 (7%)	>0.99
Rescue antiemetic use	1	5	0.19
Severity of nausea	21/3/2/4	21/4/4/1	0.46
(0/1/2/3)			
Pentazocine (mg)	0 (0-15)	0 (0-0)	0.14
Buprenorphine (mg)	0 (0-0)	0 (0-0)	0.49
24 h			
PONV	5 (17%)	5 (17%)	>0.99
Vomiting	1 (3%)	0 (0%)	>0.99
Rescue antiemetic use	0	1	>0.99
Severity of nausea	25/3/2/0	25/4/1/0	0.79
(0/1/2/3)			
Pentazocine (mg)	0 (0-0)	0 (0-0)	>0.99
Buprenorphine (mg)	0 (0-0)	0 (0-0)	0.49

Data are presented as number of patients (percentile) or median (interquartile range). PONV = postoperative nausea and vomiting. Severity of nausea: 0 = no of nausea, 1 = mild nausea, 2 = moderate nausea, 3 = severe nausea.

## Data Availability

The datasets used and/or analyzed in the current study are available from the corresponding author upon reasonable request.

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
