# Peer review of "Remimazolam’s Effects on Postoperative Nausea and Vomiting Are Similar to Those of Propofol after Laparoscopic Gynecological Surgery: A Randomized Controlled Trial"

_jcm, 2023, doi:10.3390/jcm12165402_

Round 1
Reviewer 1 Report
The authors tried to compare the effects of remimazolam and propofol in preventing PONV. But the authors used sevoflurane's data to calculate the sample size. This is inappropriate.
Minor editing of English language required.
Author Response
Response to Reviewer 1 Comments
The authors tried to compare the effects of remimazolam and propofol in preventing PONV. But the authors used sevoflurane's data to calculate the sample size. This is inappropriate.
Response: As you pointed out, we calculate the sample size based on the data comparing the incidence of PONV between sevoflurane and propofol because remimazolam is a new intravenous anesthetic approved in Japan in 2020 for general anesthesia for the first time in the world and there was no data on PONV caused by total intravenous anesthesia except for propofol at that time.

Reviewer 2 Report
This study compares the prophylactic effects on PONV between remimazolam and propofol in gynecololgic patients undergoing laparoscopic surgery.
The manuscript were neatly written and showed clear results.
Followings are some minor points that should be addressed.
- Was flumazenil used in REM group during emergence from anesthesia?
If so, could flumazenil have influenced the occurrence of PONV?
- In the current study, peripheral nerve block was used for postoperative analgesia instead of opioid-based patient-controlled analgesia. Do you think that the above fact could have caused the low incidence of PONV (17-30%) in these high-risk patients?
- P2L77 : BISTM --> BISTM
- P6L212 : masked the development of --> may have masked the difference between
Author Response
Response to Reviewer 2 Comments
Was flumazenil used in REM group during emergence from anesthesia?
If so, could flumazenil have influenced the occurrence of PONV?
Response: Flumazenil could affect the occurrence of PONV. Therefore, we did not administer flumazenil in this study. We added the sentence. (P3L104)
In the current study, peripheral nerve block was used for postoperative analgesia instead of opioid-based patient-controlled analgesia. Do you think that the above fact could have caused the low incidence of PONV (17-30%) in these high-risk patients?
Response: The choice of peripheral nerve block is one of the reasons for the low incidence of PONV. In this study, we conducted total intravenous anesthesia without fentanyl not only after the operation but also during the surgery. In addition, we administered ondansetron for the prevention of PONV. We think these approaches may also result in a low occurrence of PONV in high-risk patients.
P2L77 : BISTM --> BISTM
P6L212 : masked the development of --> may have masked the difference between
Response: Thank you for the kind indications. We corrected the words. (P2L87, P7L218)

Reviewer 3 Report
The objective of this study was to compare propofol and remimazolam for preventing PONV in laparoscopic gynecologic surgery. The authors hypothesized that propofol was more effective than remimazolam in preventing PONV in laparoscopic gynecologic surgery.The introduction section is relatively short. I would encourage the authors to elaborate on the negative consequences of postoperative nausea and vomiting. Please also expand the description of the mechanism of action and pharmacological effects of Remimazolam. Methods: "The current two-center prospective study" - Please specify the study design. Please state that it was an RCT. Conclusions: The authors wrote that "This suggests similar anti-emetic effects between both medications, but further investigation is necessary to identify the differences between remimazolam and propofol in the incidence of PONV after laparoscopic gynecological surgery during the postoperative period". I would suggest you to give some recommendations on how further studies can identify the differences and why they are needed if your study has not found any differences. Please suggest what changes to your study design can be made in future RCTs. You may write it at the end of the discussion section. Would you recommend increasing the sample size in future studies? The sample size for this study was calculated based on the incidence of PONV after general anesthesia with sevoflurane and propofol anesthesia. Since the incidence of PONV is much higher when volatile anesthetics are used )compared with propofol), probably in the studies comparing propofol with remimazolam the sample size should be larger. It is also written, "Additional studies are needed to investigate the impact of remimazolam on PONV in patient groups with various backgrounds". Could you please specify what kind of backgrounds? Although this manuscript is relatively well written, some additional English-language editing is recommended to improve the quality of the manuscript.
Some additional English-language editing is recommended to improve the quality of the manuscript.
Author Response
Response to Reviewer 3 Comments
The introduction section is relatively short. I would encourage the authors to elaborate on the negative consequences of postoperative nausea and vomiting. Please also expand the description of the mechanism of action and pharmacological effects of Remimazolam.
Response: Thank you for the suggestion. We added extra sentences about the negative consequences of PONV and the mechanism of action and pharmacological effects of remimazolam. (P1L37-42, P251-56)
Methods: "The current two-center prospective study" - Please specify the study design. Please state that it was an RCT.
Response: We corrected the sentence following your indication. (P2L68)
Conclusions: The authors wrote that "This suggests similar anti-emetic effects between both medications, but further investigation is necessary to identify the differences between remimazolam and propofol in the incidence of PONV after laparoscopic gynecological surgery during the postoperative period". I would suggest you to give some recommendations on how further studies can identify the differences and why they are needed if your study has not found any differences. Please suggest what changes to your study design can be made in future RCTs. You may write it at the end of the discussion section. Would you recommend increasing the sample size in future studies? The sample size for this study was calculated based on the incidence of PONV after general anesthesia with sevoflurane and propofol anesthesia. Since the incidence of PONV is much higher when volatile anesthetics are used) compared with propofol), probably in the studies comparing propofol with remimazolam the sample size should be larger. It is also written, "Additional studies are needed to investigate the impact of remimazolam on PONV in patient groups with various backgrounds". Could you please specify what kind of backgrounds?
Response: In this study, there was no significant difference between the two groups. This might be due to the limited sample size. As you pointed out, we calculate the sample size based on the data comparing the incidence of PONV between sevoflurane and propofol because remimazolam is a new intravenous anesthetic approved in Japan in 2020 for general anesthesia for the first time in the world and there was no data on PONV caused by total intravenous anesthesia except for propofol at that time. Therefore, we think further study is necessary to investigate the effect of remimazolam on PONV with a larger sample size. As for patients’ backgrounds, we included females only. In addition, the incidence of PONV is influenced by the type of surgery. For the above reasons, we revised the sentences in the discussion and conclusion sections. (P7L215-226)
Although this manuscript is relatively well written, some additional English-language editing is recommended to improve the quality of the manuscript.
Response: We used an English-language editing service to improve the quality of the manuscript.

Round 2
Reviewer 3 Report
Dear Authors,
The manuscript was improved and the comments were addressed.
I do not have any further comments.